# Comparative Transcriptome Analyses of Different *Rheum officinale* Tissues Reveal Differentially Expressed Genes Associated with Anthraquinone, Catechin, and Gallic Acid Biosynthesis

**DOI:** 10.3390/genes13091592

**Published:** 2022-09-05

**Authors:** Lipan Zhou, Jiangyan Sun, Tianyi Zhang, Yadi Tang, Jie Liu, Chenxi Gao, Yunyan Zhai, Yanbing Guo, Li Feng, Xinxin Zhang, Tao Zhou, Xumei Wang

**Affiliations:** School of Pharmacy, Xi’an Jiaotong University, Xi’an 710061, China

**Keywords:** *Rheum officinale*, transcriptome, anthraquinone, biosynthesis, SSR, differentially expressed genes

## Abstract

*Rheum officinale* Baill. is an important traditional Chinese medicinal herb, its dried roots and rhizomes being widely utilized to cure diverse diseases. However, previous studies mainly focused on the active compounds and their pharmacological effects, and the molecular mechanism underlying the biosynthesis of these ingredients in *R. officinale* is still elusive. Here, we performed comparative transcriptome analyses to elucidate the differentially expressed genes (DEGs) in the root, stem, and leaf of *R. officinale*. A total of 236,031 unigenes with N50 of 769 bp was generated, 136,329 (57.76%) of which were annotated. A total of 5884 DEGs was identified after the comparative analyses of different tissues; 175 and 126 key enzyme genes with tissue-specific expression were found in the anthraquinone, catechin/gallic acid biosynthetic pathway, respectively, and some of these key enzyme genes were verified by qRT-PCR. The phylogeny of the *PKS III* family in Polygonaceae indicated that probably only *PL_741 PKSIII1*, *PL_11549 PKSIII5*, and *PL_101745 PKSIII6* encoded *PKSIII* in the polyketide pathway. These results will shed light on the molecular basis of the tissue-specific accumulation and regulation of secondary metabolites in *R. officinale*, and lay a foundation for the future genetic diversity, molecular assisted breeding, and germplasm resource improvement of this essential medicinal plant.

## 1. Introduction

*Rheum officinale* Baill. is an essential medical plant that belongs to the genus *Rheum* within Polygonaceae, which has been formally included in the Chinese Pharmacopoeia as one of the resource plants of rhubarb (*Da-Huang* in Chinese), a traditional medicine in China. Its dried roots and rhizomes have been extensively used in many countries with various therapeutic effects such as clearing body heat, cooling blood and detoxifying toxins, relieving dampness, and abating jaundice [1,2,3]. Numerous studies have indicated that rhubarb contains various chemical ingredients, and anthraquinones which make up the major bioactive composition with multiple pharmacological effects [3,4,5]. Especially, Rolta et al. [6] reported emodin can inhibit the spike protein of SARS-CoV-2 binding to the ACE2 receptor for the treatment of COVID-19. However, research about the candidate genes for the biosynthesis of effective ingredients in *R. officinale* is limited, and the molecular mechanism of the biosynthesis in different tissues is still poorly understood due to inadequate genomic information.

In embryophytes, the biosynthesis of anthraquinones is principally involved in the upstream pathways consisting of the shikimate pathway, methyl-D-erythritol 4-phosphate (MEP) pathway, mevalonate (MVA) pathway, polyketide pathway, and the downstream pathways for anthraquinone modification derivatized by UDP-glycosyltransferases (UGTs) and Cytochrome P450s (CYPs) genes [7]. Nevertheless, a previous study indicated that the synthesis of emodin-type anthraquinone is predominantly through the polyketide pathway, while the *Rubia*-type anthraquinones are derived from the shikimate pathway [8]. Moreover, the biosynthesis of catechin and gallic acid of our concern is a complex process involving the shikimate pathway, phenylpropanoid biosynthetic pathway, and flavonoid biosynthetic pathway [9]. Secondary metabolites are usually differentially distributed in diverse tissues of higher plants, which are associated with the tissue-specific expression of biosynthetic enzyme genes and transcript factors (TFs) related to the corresponding biosynthetic pathways [10,11,12]. Therefore, elaborating the expression differences of key enzyme genes in different tissues would deepen our understanding of the molecular mechanisms underlying the tissue specificity of the active compounds [5,13,14].

Currently, RNA sequencing (RNA-seq) is broadly used as an important approach for gene expression level analyses and molecular marker development [7,12,13,15]. Many studies have utilized transcriptomic analyses to identify critical candidate genes and tissue-specific differentially expressed genes (DEGs) for the biosynthesis of terpenoid, stilbene, saponin, lignin, anthraquinone, and flavonoid [11,16,17]. Zhou et al. [7] have reported the candidate genes associated with anthraquinone biosynthesis and their tissue-specific expression patterns in *Rheum tanguticum*. Liu et al. [18] combined transcriptomic and metabolomic analyses to further reveal the differences in the expression of anthraquinones and flavonoids between two *Rheum* species. Although previous studies have confirmed the differences in the accumulation of secondary metabolites in different tissues of *R. officinale* and used RNA-seq to identify candidate genes involved in the biosynthesis of anthraquinones, etc. [16,19,20,21,22], the differences in the expression of the enzyme genes associated with the synthesis of these components are still indefinite in *R. officinale*.

In this work, we generated a comprehensive transcriptome for *R. officinale* and identified the candidate genes associated with anthraquinone, catechin, and gallic acid biosynthesis. Furthermore, comparative transcriptome analyses were carried out to compare and elucidate the expression profiles in *R. officinale* root, stem, and leaf tissues. The results could be valuable genetic resources for providing comprehensive insights into molecular mechanisms of tissue-specific distribution of active components and improving the quality and production of this essential medicinal plant.

## 2. Materials and Methods

### 2.1. Plant Materials

Root, leaf, and stem samples of *R. officinale* were gathered from the three individuals as biological replicates at the time of blossoming in Pingli County, Shaanxi Province, China (32°01′ N, 109°21′ E). A total of nine samples were prepared for transcriptome sequencing. All the samples were immediately frozen in liquid nitrogen and stored at −80 °C before RNA extraction.

### 2.2. RNA Extraction, cDNA Library Construction, and Illumina Sequencing

The RNA in each sample was extracted utilizing the E.Z.N.A.^®^ Plant RNA Kit (Omega Bio-Tek, Norcross, GA, USA). A Qubit 2.0 Fluorometer (Life Technologies, Foster City, CA, USA) was applied to measure RNA concentration. RNA degradation, contamination and integrity were determined by the Agilent 2100 Bioanalyzer (Agilent Technologies, Palo Alto, CA, USA) and 2% agarose gels. The purity of RNA was detected with a NanoDrop ND-2000 spectrophotometer (NanoDrop products, Wilmington, DE, USA). The mRNA was enriched using the NEBNext Poly(A) mRNA Magnetic Isolation Module (E7490, NEB, Ipswich, UK) from 5 µg of total RNA, and sequencing libraries were prepared using NEBNext mRNA Library Prep Master Mix Set for Illumina (E6110, NEB) and NEBNext Multiplex Oligos for Illumina (E7500, NEB). Library insert sizes range from 100 to 200 bp. The insertion fragment sizes of prepared libraries were resolved on 1.8% agarose gel. Finally, size selected libraries were quantified using the Library Quantification Kit-Illumina GA Universal (KK4824, Kapa, Wilmington, DE, USA). The qualified libraries were amplified by bridge PCR to generate clustered template DNA fragments on the Illumina cbot [23], then paired-end sequencing with the length of 150 bp was performed to generate raw data for nine samples by the Illumina Hiseq X Ten platform.

### 2.3. De Novo Assembly and Functional Annotation

To obtain high-quality clean reads, all the raw reads of *R. officinale* transcriptome generated by Illumina sequencing were processed through the removal of the adaptor, artificial primers, read with uncertain bases “N” over 5%, and low-quality sequences using Trimmomatic v0.35 [24]. After trimming, all clean reads were de novo assembled using Trinity v2.13.2 with the default parameters to generate transcripts [25]. Furthermore, such assembled transcripts were clustered to remove redundancy using CD-HIT-EST v.4.8.1 with a 95% identity and 95% query coverage (for the shorter sequence) threshold, and the remaining unigenes were then used for the subsequent analyses [26].

To predict the potential functions of assembled unigenes, all non-redundant unigenes were searched against public databases including Nr (https://www.ncbi.nlm.nih.gov/refseq/, accessed on 7 October 2021), Swiss-Prot (https://www.uniprot.org/, accessed on 7 October 2021), COG (https://www.ncbi.nlm.nih.gov/research/cog/, accessed on 10 October 2021), KOG (https://ftp.ncbi.nih.gov/pub/COG/KOG/, accessed on 12 October 2021), and TrEMBL (http://www.bioinfo.pte.hu/more/TrEMBL.htm, accessed on 21 October 2021) using BLASTX (version 2.13.0) (https://blast.ncbi.nlm.nih.gov/, accessed on 8 October 2021) with an e-value cutoff of 10^−5^. According to the Nr annotation results, the functional annotation information of GO (http://www.geneontology.org/, accessed on 5 November 2021) was implemented on Blast2GO v2.5 (https://www.blast2go.com/, accessed on 7 November 2021) with an e-value threshold of 10^−5^. The KEGG (https://www.kegg.jp/, accessed on 13 November 2021) automated annotation server KASS (https://www.genome.jp/tools/kaas/, accessed on 14 November 2021) was assigned to designate KEGG classification of assembled unigenes with a threshold of 10^−10^. All nucleotide sequences of the unigenes were searched against the Pfam database (http://ftp.ebi.ac.uk/pub/databases/Pfam/releases/Pfam34.0/, accessed on 27 November 2021) using HMMER v3.3.2 (e-value < 10^−5^) [27].

### 2.4. Phylogenetic Analyses of Type III Polyketide Synthases (PKS III) Genes in R. officinale

To identify the *PKS III* genes in *R. officinale*, 145 protein sequences belonging to *PKS III* were selected and downloaded from NCBI (https://www.ncbi.nlm.nih.gov/, accessed on 20 November 2021) (Appendix A). The Getorf program (https://emboss.bioinformatics.nl/cgi-bin/emboss/getorf, accessed on 23 November 2021) was used to detect open reading frames (ORFs) containing at least 150 amino acids for the unigenes of *R. officinale*, and then we used BLASTP to query 147 protein sequences with the ORFs with the e-value threshold of 10^−5^. Seven unigenes that displayed high similarity to *PKS III* were identified and used in the subsequent analyses. To infer the phylogenetic relationships of the *PKS III* family in Polygonaceae, 50 protein sequences of *PKS III* downloaded from NCBI (Appendix A) and 7 protein sequences detected from *R. officinale* were prepared, and the bacterial *PKS III* Mycobacterium tuberculosis *PKS10* (CAB06631.1) was used as the outgroup. Alignment of all protein sequences was conducted using the MUSCLE algorithm in MEGA-X, and then alignment was trimmed with trimAL. Finally, a maximum likelihood (ML) phylogenetic tree was constructed using MEGA-X with the JTT + G model selected by modeltest, and the bootstrap replicate was set to 1000 [28].

### 2.5. Differential Gene Expression Analyses

Clean reads of nine *R. officinale* samples were matched to the unigenes through Bowtie v2.4.4 software, and quantification of gene expression level was performed using RSEM v1.3.3 with default parameters to assess the expression abundance of all unigenes based on FPKM (fragments per kilobase of transcript per million mapped reads) [29,30]. Analyses of DEGs between two samples were conducted by DESeq2 package in RStudio and set thresholds with |log_2_(foldchange)| ≥1 and false discovery rate (FDR) ≤ 0.05 (Control/Treated) to evaluate the statistically significant level [31]. In three comparisons of the root, stem, and leaf tissues in *R. officinale*, unigenes having foldchange (FC) value greater than 2 were regarded as up-regulated, while those less than 0.5 were down-regulated. DEGs of different tissues were also functionally annotated with the aforementioned eight public databases to elaborate the functions and metabolic pathways. Afterward, GO and KEGG enrichment analyses of DEGs were carried out with ClusterProfile package [32]. The FPKM values were log-transformed and normalized using the Pheatmap package to draw a heatmap describing transcript abundance levels.

### 2.6. Transcription Factor Analysis

The predicted longest ORF of each sequence was considered as the potential coding region sequence (CDS). The CDS of DEGs of *R. officinale* roots, stems, and leaves were used to predict the putative TF families through the Transcription Factor Prediction (http://planttfdb.gao-lab.org/prediction.php, accessed on 3 November 2021) in Plant Transcription Factor Database (PlantTFDB).

### 2.7. Quantitative Real-Time PCR Verification

qRT-PCR was conducted for 16 DEGs related to the anthraquinone, catechin and gallic acid biosynthetic pathways to validate the reliability of the RNA-seq. We used NCBI Primer-BLAST (https://www.ncbi.nlm.nih.gov/tools/primer-blast/, accessed on 20 December 2021) to design and validate the primers (Appendix A). The total RNA of each sample was extracted using the E.Z.N.A.^®^ Plant RNA Kit (Omega Bio-Tek). Then, the extracted RNA was removed from the genomic DNA using DNase (Tiangen, Beijing, China), and the cDNA was synthesized using Thermo Scientific RevertAid First Strand cDNA Synthesis Kit (Thermo Fisher, Beijing, China). A 20 μL qRT-PCR reaction volume (10 μL of 2× NovoStart^®^SYBR qPCR SuperMix Plus, 0.4 μL of each forward and reverse primers, 2 μL of cDNA template, and 7.2 μL of RNase Free water) was prepared with the NovoStart^®^ SYBR qPCR SuperMix Plus (Novoprotein, Shanghai, China), and three technical replicates were conducted for each sample to guarantee reliability. Next, all the reactions were further performed with BIO-RAD CFX Connect Real-Time PCR detect System (BIORAD, Hercules, CA, USA) based on the following protocol: 95.0 °C for 60 s (initial denaturation), followed by 42 cycles of 95.0 °C for 20 s (denaturation), 56.0 °C for 20 s (annealing), 72.0 °C for 30 s (extension), and melt curve: 97.0 °C for 10 s, 65.0 °C for 30 s, 95.0 °C for 30 s. Eventually, the housekeeping gene *actin* was normalized to other genes and relative expression levels of genes were determined based on the 2^−ΔΔCt^ method [33].

### 2.8. Identification of Simple Sequence Repeat (SSR)

The candidate SSRs of all the non-redundant *R. officinale* unigenes were detected by MISA software with parameters set to dinucleotide, trinucleotide, tetranucleotide, pentanucleotide, and hexanucleotide, and the motifs minimum repeats were 6, 4, 3, 3, 2, respectively [34]. In this study, mononucleotide repeats were excluded due to possible sequencing errors, mismatch, and the difficulty of distinguishing this SSR type on a polypropylene electrophoresis gel. In addition, we used the program Primer 3 to select SSRs with appropriate flanking lengths to design PCR primers (Appendix A). The following criteria were considered for designing the primers: primer length of 18–23 nucleotides; PCR product size range of 100 to 300 bp; GC content of 30–70% and annealing temperature between 50 and 70 °C with 55 °C as the optimum melting temperature.

## 3. Results

### 3.1. Subsection RNA Sequencing and De Novo Assembly

The nine cDNA libraries from the roots, stems, and leaves of *R. officinale* were named Ro_R1, Ro_R2, Ro_R3, Ro_S1, Ro_S2, Ro_S3, Ro_L1, Ro_L2, Ro_L3, respectively, and sequenced via the Illumina Hiseq X Ten platform. After sequencing and quality control, we obtained libraries with clean reads ranging from 24,265,150 to 32,077,349, GC content ranging from 48.61 to 50.72%, and Q30 values up to 94.26% (Appendix A). These high-quality reads were available for the subsequent correlation analyses.

After de novo assembly, a sum of 463,056 transcripts with an average sequence length of 839.83 bp and N50 of 1675 bp was generated. Then CD-HIT-EST was used to remove redundant parts of these transcripts, resulting in a total of 236,031 unigenes with an average length of 563.98 bp, a GC content of 45.83%, and N50 of 769 bp for the following analyses (Table 1). The length distribution of the transcripts and unigenes is demonstrated in Appendix A, where it can be seen that the unigene length of *R. officinale* was mainly distributed between 200 and 400 bp.

### 3.2. Functional Annotation

The statistics of the number of best BLAST hits for unigenes in each database are shown in Table 2, a total of 126,539 (53.61%), 41,568 (17.61%), 74,696 (31.65%), 20,133 (8.53%), 36,789 (15.59%), 56,876 (24.10%), 75,256 (31.88%), 102,507 (43.43%) of the *R. officinale* unigenes was annotated against the Nr, COG, KOG, GO, KEGG, Swiss-Prot, Pfam and TrEMBL databases, respectively. Overall, 136,329 (57.76%) of the assembled unigenes were found to be homologs of at least one BLAST hit in the eight databases. In addition, the Venn diagram (Figure 1A) indicated that a total of 4277 (1.81%) unigenes could be annotated in the Nr, COG, GO, KEGG, and Swiss-Prot databases. Nr annotation results showed the greatest homology with *R. officinale* was *Beta vulgaris* (6994 unigenes, 5.53%), followed by *Rhodosporidium toruloides* (6564, 5.19%), *Phaeosphaeria nodorum* (4592, 3.63%), *Marssonina brunnea* (3480, 2.75%), *Vitis vinifera* (3438, 2.72%), *Leptosphaeria maculans* (2928, 2.31%), *Microbotryum violaceum* (2793, 2.21%), *Auricularia delicata* (2247, 1.78%), *Glarea lozoyensis* (2013, 1.59%), and *Oidiodendron maius* (1686, 1.33%), in addition, the rest of the 70.92% unigenes were matched to other plants (Figure 1B).

For COG annotation, 41,568 annotated unigenes were classified into 25 categories, with a high proportion of “posttranslational modification, protein turnover, chaperones” (2596, 6.25%), “translation, ribosomal structure and biogenesis” (2267, 5.45%), and “carbohydrate transport and metabolism” (1927, 4.64%) (Appendix A). Then, 74,696 unigenes were divided into 25 KOG categories based on the biological functions of their orthologous proteins. The most prominent of these classifications was “general functional prediction” (6922, 9.27%), followed by “posttranslational modification, protein turnover, chaperones” (4880, 6.53%), “translation, ribosomal structure and biogenesis” (3430, 4.59%), and “signal transduction mechanisms” (2756, 3.69%) (Figure 1C).

GO annotation revealed that 20,133 unigenes were successfully classified and categorized into three major categories, including 14 subgroups for molecular function (MF), 20 subgroups for biological process (BP), and 12 subgroups for cellular component (CC). Depending on sequence homology, “catalytic activity” (10,917 unigenes, 54.22%), “metabolic processes” (13,840 unigenes, 68.74%), and “cell part” (5996 unigenes, 29.78%) represented the most abundant terms in MF, BP, and CC, respectively (Figure 1D). A total of 36,789 unigenes annotated in the KEGG database was mapped into the following five categories: “metabolism” (9878 unigenes), “genetic information processing” (9516 unigenes), “cellular processes” (2086 unigenes), “environmental information processing” (550 unigenes), and “organismal systems” (412 unigenes). Among the subcategories, “carbohydrate metabolism” (ko01200, 2190 unigenes) and “ribosome” (ko03010, 1896 unigenes) were predominant (Appendix AB). Furthermore, 9878 unigenes from 96 metabolism pathways were further counted, and a total of 856 unigenes was involved in secondary metabolic pathways such as phenylpropanoid biosynthesis, terpenoid biosynthesis, flavonoid biosynthesis, tropane, piperidine, and pyridine alkaloid biosynthesis. Among them, “phenylpropanoid biosynthesis” (ko00940) and “terpenoid backbone biosynthesis” (ko00900) were dominant, which contained 272 unigenes belonging to *PAL*, *TAL*, *C4H*, *4CL*, etc. and 202 unigenes belonging to *GGPPS*, *FPS*, *G10H*, *IS*, etc., respectively (Appendix A).

### 3.3. Differentially Expressed Gene Analyses between Different Tissues

The overall quality assessment of gene expression in the nine samples using FPKM values showed that the correlation coefficients of samples between biological replicates of *R. officinale* were greater than those outside the biological replicates. Among the 3 biological replicates of roots: the correlation coefficients between Ro_R1 and Ro_R2, Ro_R2 and Ro_R3, Ro_R1 and Ro_R3 were 0.66, 0.70, and 0.92, respectively, smaller than the coefficients between replicates from stems (over 0.74) and leaves (over 0.75), which indicated that the differences of gene expression in roots were higher than those in stems and leaves (Appendix A).

A total of 261 DEGs was commonly present between paired comparisons (Ro_R vs. Ro_L, Ro_R vs. Ro_S, and Ro_S vs. Ro_L) (Figure 2A). Moreover, the pairwise comparisons of Ro_R vs. Ro_L, Ro_R vs. Ro_S, and Ro_S vs. Ro_L (control vs. treated) resulted in a total of 3641 DEGs (54.44% up-regulated and 45.56% down-regulated), 3308 DEGs (55.44% up-regulated and 44.46% down-regulated), and 2380 DEGs (43.03% up-regulated and 56.97% down-regulated), respectively (Figure 2B). All 5884 DEGs obtained from each comparison group of three different tissues were screened for hierarchical clustering analysis, and the heatmap of the DEGs showed that the gene expression profiles of roots, stems, and leaves of *R. officinale* were similar (Figure 2C).

We found that 3362, 2984, and 2118 DEGs in Ro_R vs. Ro_L, Ro_R vs. Ro_S, and Ro_S vs. Ro_L could be annotated against the Nr, COG, KOG, GO, KEGG, Swiss-Prot, Pfam, and TrEMBL databases, respectively. We further performed GO and KEGG enrichment analyses of these DEGs in different tissues. In the GO enrichment analysis, the enriched items of all paired comparison groups were “embryo development ending in seed dormancy” (GO:0009793) for the BP category, and “chloroplast” (GO:0009507) and “chloroplast stroma” (GO:0009570) in the CC category. In the MF category, “xyloglucan:xyloglucosyl transferase activity” (GO:0016762) was significantly enriched in Ro_R vs. Ro_L and Ro_R vs. Ro_S, and “sucrose synthase activity” (GO:0016157) was enriched in Ro_S vs. Ro_L (Appendix A). KEGG enrichment results demonstrated that the enriched terms for Ro_R vs. Ro_L, Ro_R vs. Ro_S, and Ro_S vs. Ro_L were “starch and sucrose metabolism”, “ribosome biogenesis in eukaryotes”, and “biosynthesis of amino acids”, respectively (Appendix A). Further statistics of the secondary metabolic pathways in the KEGG enrichment analysis revealed that phenylpropanoid biosynthesis was the most abundant in the three comparison groups. In addition, flavonoid biosynthesis, carotenoid biosynthesis, isoquinoline alkaloid biosynthesis, ubiquinone, and other terpenoid-quinone biosyntheses were present (Appendix A).

### 3.4. Transcription Factors (TFs) Analysis of DEGs

By conducting TF prediction for the DEGs in the three comparison groups, we found that 69, 68, and 64 DEGs could be predicted for Ro_R vs. Ro_L, Ro_R vs. Ro_S, and Ro_S vs. Ro_L, respectively, and further classified into 27, 25, and 25 TF families, respectively (Appendix A). Meanwhile, 2 DEGs and 14 TF families were shared among the three comparison groups (Figure 3A,B). Among them, the ethylene-responsive factor (ERF, 26 DEGs) TF family was the most predominant, followed by the basic helix-loop-helix (bHLH, 21 DEGs), C2H2(16 DEGs), basic leucine zipper (bZIP, 14 DEGs), and myeloblastosis (MYB, 10 DEGs) TF families (Figure 3C). Notably, the bHLH family (11 DEGs) was mainly present in Ro_R vs. Ro_L, while the ERF family (10 DEGs) was the most prevailing in Ro_R vs. Ro_S (Figure 3D).

### 3.5. Identification of Genes Associated with the Anthraquinone Biosynthesis

In this study, we identified 175 structural enzyme unigenes potentially regulating the biosynthesis of anthraquinone in the shikimate pathway, MEP pathway, MVA pathway, and polyketide pathway based on homology search and functional annotation (Figure 4A). In total, 50 unigenes encoding 11 structural enzymes were screened in the shikimate pathway. The expression analysis showed that *DAHPS* and *EPSPs* were expressed in 3 tissues, *DHQS* was highly expressed in stems and leaves, *DHQD/SDH*, *SK*, *CS*, and *MenE* were mainly expressed in stems, while *ICS*, *MenC*, *MenB* had higher expression levels in roots. There were 25 unigenes encoding seven enzymes of the MEP pathway. Among them, *CMS* and *CMK* tended to be highly expressed in roots; *DXR*, *CMK,* and *HDS* were highly expressed in stems; while *DXS*, *HDR,* and *HDS* were mainly expressed in leaves. We also found that the expression patterns of the 93 unigenes encoding seven enzymes of the MVA pathway were similar, all being highly expressed in roots and stems (Figure 4A and Appendix A). Interestingly, the volcano plot demonstrated the up-regulated and down-regulated unigenes in the three comparative groups of anthraquinone biosynthetic pathways. Of these genes, *MenE* was shown to be up-regulated in all three comparative groups; *IDI3* and *HMGR11* were up-regulated in Ro_R vs. Ro_L and Ro_R vs. Ro_S; *DAHPS1* was up-regulated in Ro_R vs. Ro_L and Ro_S vs. Ro_L; *HMGR3* was up-regulated in Ro_R vs. Ro_S; and *DHQD/SDH* was down-regulated in Ro_S vs. Ro_L (Figure 4B–D). In addition, we identified seven candidate genes probably encoding *PKSIII* in the polyketide pathway (Appendix A). The results of the phylogenetic tree revealed that only *PL_741 PKSIII1*, *PL_11549 PKSIII5,* and *PL_101745 PKSIII6* were clustered in the *CHS* group, while the other four unigenes (*PL_2139 PKSIII2*, *PL_3171 PKSIII3*, *PL_50026 PKSIII4,* and *PL_182516 PKSIII7*) were classified to the non-*CHS* group (Figure 5). Finally, we also found that 112 unigenes were predicted to be CYP family members and 2 unigenes could encode UGT enzymes (Appendix A).

### 3.6. Catechin and Gallic Acid Biosynthesis in R. officinale

Here, we detected a total of 126 structural enzyme unigenes for the biosynthetic pathway of catechin and gallic acid in *R. officinale*, of which nine encoding *HCT*, seven encoding *CHS*, three encoding *C3′H*, two encoding *F3H* and *F3′H*, and one encoding *F3′5′H*, *DFR*, *LAR* were identified (Appendix A). Interestingly, *DAHPS* expressed in roots, stems, and leaves, *HCT* and *CHS* showed high expression levels in roots, while the rest of the unigenes were mainly expressed in stems and leaves (Figure 6A). Furthermore, we also specifically focused on DEGs involved in catechin and gallic acid biosynthesis in three comparison groups and found the number of DEGs between Ro_S vs. Ro_L (12 DEGs) was more than that of Ro_R vs. Ro_L (5 DEGs), and Ro_R vs. Ro_S (4 DEGs) (Appendix A). In Ro_R vs. Ro_L, *DAHPS1*, *CHS4*, and *F3′5′H* were up-regulated in leaf tissues, while the expression of *CHS7* and *HCT5* was higher in roots than in the leaves (Figure 6B and Appendix A). *PAL**3*, *HCT9*, and *C3′H2* were up-regulated and *CHS7* was down-regulated in Ro_R vs. Ro_S (Figure 6C and Appendix A). For Ro_S vs. Ro_L, *PAL3*, *CHS7*, *HCT9*, and *C3′H2* were found to express more highly in stems, while the expression of the remaining eight unigenes (*DAHPS1*, *CHS4*, *F3′5′H*, *F3′H2*, *DHQD/SDH9*, *F3H1*, *LAR*, *ANS2*) were up-regulated in leaves (Figure 6D and Appendix A).

### 3.7. Quantitative Real-Time PCR (qRT-PCR) Validation

To further check the reliability of the RNA-seq results, qRT-PCR validation was conducted for the 16 randomly-selected DEGs (*IDI3*, *HMGR11*, *DHQD/SDH9*, *PAL3*, *CHS4*, *HCT6*, *C3′H2*, *F3′H2*, *F3H1*, *DFR*, *ANS2*, *MenB1*, *CMK*, *PMK3*, *PL_741 PKS III1*, *PL_11549 PKS III5*) involved in the biosynthesis of the anthraquinone, catechin and gallic acid biosynthetic pathway. Although Log_2_(FPKM + 1) derived from RNA-seq analysis and the relative expression levels of qRT-PCR analysis in roots, stems, and leaves of *R. officinale* were not exactly analogous, the expression trends of both showed concordance (Appendix A).

### 3.8. Detection of SSR Loci

We identified a total of 127,612 SSR loci in 236,031 unigene sequences of *R. officinale* using MISA, and 26,674 unigenes contained at least two SSR sites. The SSRs in the transcriptome of *R. officinale* were abundant, and all repeat types from dinucleotide to hexanucleotide were present, with the largest number of hexanucleotide repeats accounting for 74.24% of the total SSRs, followed by trinucleotides (18,635, 14.6%), and the remaining four nucleotide repeat types were relatively few, accounting for only 11.16% (Appendix A). In addition, the count of repeat motifs was mainly 2 to 5, which accounted for 95.06% of the total, followed by 6–10 repeats (4.54%) mainly distributed in tri- and hexanucleotide repeats, and more than 15 repeats were primarily observed in dinucleotide repeat types (Appendix A). Among the trinucleotide repeats, AGG/CCT, with 3165 SSRs (16.98%), was the most dominant repeat motif, while ACT/AGT (1.44%) occupied the least. The most frequent hexanucleotide repeat motif was AAAAAG/CTTTTT with the number of 2086, accounting for 2.14% (Appendix A).

## 4. Discussion

### 4.1. De Novo Assembly and Functional Annotation

Following the emergence of next-generation sequencing (NGS) technology, RNA-seq has become a central and powerful tool for studying metabolites of interest in non-model species, with the advantage of high throughput and relative rapidity [35,36,37]. RNA-seq can be an effective method to study gene function annotation, the discovery of novel biosynthetic enzyme genes, differential distribution of secondary metabolites in diverse tissues, gene expression levels, gene regulation, environmental factors, and gene expression networks [13,37,38]. Here, we report 236,031 assembled unigenes with N50 of 769 bp from diverse tissues (i.e., roots, stems, and leaves) of nine *R. officinale* samples. This result far exceeds the gene database of the transcriptome of *R. palmatum* seedlings [39] but is similar to the genes yielded by high-throughput sequencing in the roots, stems, and leaves of *R. tanguticum* [7], which may be related to different sampling periods and immature tissues in seedlings containing limited transcriptome information.

After sequencing and assembly, functional annotation is one of the most important steps, which can offer relevant biological insights into genomic and transcriptomic data through homology alignment [40]. For example, DEGs screened from a certain secondary metabolite biosynthetic pathway in different tissues are functionally annotated to reveal the molecular mechanism of tissue-specific distribution of metabolites [11,13,41]. We found a total of 136,329 (57.76%) of the assembled unigenes could be successfully annotated in eight public databases, indicating that there are many unigenes with unknown sequence characteristics and functions in the transcriptome of *R. officinale*. Wang et al. [11] obtained a total of 80,981 unigenes from the root, stem, and leaf tissues in *Polygonum cuspidatum*, a related species to *R. officinale*, and 40,729 of them (50.29%) were annotated. The result may be due to the scarcity of reference genomic resources for *Rheum* or even in the Polygonaceae. In brief, the assembly and functional annotation of *R. officinale* are broadly available, which will offer an abundant genetic resource for studying the molecular mechanisms of tissue-specific distribution of active secondary metabolites and aiding efficient breeding to alleviate this medicinal plant shortage.

### 4.2. Candidate Genes Identification and DEG Analyses Associated with Secondary Metabolite Biosynthesis

Anthraquinone and its derivatives are aromatic polyketides that can be widely synthesized by bacteria, fungi, lichens, insects, and plants, and have various functions such as photoprotection, improvement of plant disease resistance, as well as various medicinal effects [42]. The biosynthesis of anthraquinone mainly results from the shikimate/o-succinylbenzoic acid pathway and polyketide pathway, and the products can be modified with UGTs, CYP450s [7,42,43]. PKSs can be divided into types I, II, and III PKSs according to their structure [44]. Type III PKSs, the chalcone synthase superfamily, are mainly distributed in the plants, but have also been identified recently in bacteria and fungi [19,44,45]. At present, the mechanisms underlying most polyketide biosynthetic pathways are still elusive, but *PKS III* of plants certainly plays a key role in the initial reaction of these pathways [46,47]. The 26 unigenes found in the roots, stems, and leaves of *P. cuspidatum* could be annotated to 18 enzymes participating in the formation of the anthraquinone skeleton [11]. Here, we carried out comparative transcriptome analyses with specific attention to the expression levels of the enzyme genes in the biosynthetic pathways of anthraquinone, catechin, and gallic acid in different tissues. Combined with functional annotation, we identified 175 tissue-specific candidate genes in the anthraquinone biosynthetic pathway of *R. officinale,* and only three of the seven candidate genes that might encode *PKSIII* in the polyketide pathway (Figure 5 and Appendix A). The results of the analysis revealed that many DEGs displayed tissue-specific expression (e.g., *ICS*, *MenC*, *MenB* were highly expressed in roots; *DHQD/SDH*, *SK*, *CS*, and *MenE* were mainly expressed in stems), which implied that the accumulation levels of secondary metabolites might differ among different tissues [12]. The biosynthetic pathways of catechins and gallic acid start with the common precursors, phosphoenolpyruvate and erythrose 4-phosphate, and share part of the shikimate pathway with anthraquinone biosynthesis (Figure 4A and Figure 6A). Nonetheless, there are two synthetic pathways for gallic acid, but it is generally accepted that the short pathway directly from 3-dehydroshikimic acid is predominant. A total of 126 structural enzyme genes for the biosynthesis processes were found, among them, *DAHPS*, *HCT*, and *CHS* had high expression levels in roots, while the rest were expressed in stems and leaves (Figure 6A). Indeed, these results further demonstrate that RNA-seq is an important method for exploring key enzyme genes associated with the biosynthesis of interest secondary metabolites, enabling potential genetic improvements, enhancing the content of compounds, assisting molecular breeding, and facilitating functional gene research in rhubarb.

### 4.3. Transcription Factor Analysis

TFs are proteins that bind DNA in a sequence-specific manner and regulate transcription by acting as recruitment required for RNA polymerase during transcription initiation [16,48]. In plants, TFs can improve disease and stress resistance, mediate growth and development, affect intercellular signaling, regulate the synthesis and accumulation of secondary metabolites, as well as influence plant evolution [10,16,49,50]. Several studies have demonstrated that the MYB-bHLH-WD40 complex in plants can regulate the biosynthesis of flavonoids, changing the types and contents of metabolites [51]. In addition, NAC, ERF, C2H2, C3H, WRKYs, and MYB-related TF families can also mediate the synthesis of secondary metabolites [10,16,52]. The most abundant TF families predicted in *R. officinale* included bHLH, followed by ERF, C2H2, bZIP, and MYB (Figure 3C). Notably, the bHLH and ERF families were the most differentially expressed in Ro_R vs. Ro_L and Ro_R vs. Ro_S, respectively (Figure 3D). The bHLH and bZIP TF families mainly play crucial roles in plant growth and development, physiological metabolism, and stress response, and the ERF TF family is involved in signaling pathways such as salicylic acid, jasmonic acid, and ethylene [13,53]. However, TFs related to the biosynthesis of anthraquinones have not yet been reported, and this should be explored more in-depth in the future. The MYB TF family is necessary for the phenylpropanoid synthesis and metabolism pathway, therefore, it can regulate the biosynthetic pathway of catechins [54]. TFs identified from the *R. officinale* database may contribute in altering the content and tissue-specific accumulation of secondary metabolites as well as responding to adverse effects.

### 4.4. Simple Sequence Repeat (SSR) Analysis

With the rapid development of molecular biology-related detection technologies such as high-throughput sequencing, the use of RNA-seq expressed sequences to develop molecular markers has obvious advantages, especially for non-model species without a reference genome [20,23,55]. The developed SSRs are tandem repeat sequences consisting of 1–6 nucleotides in the genome, distributed in both coding and non-coding regions of genes, and are the most suitable markers for constructing high-throughput genotyping with co-dominant inheritance, high polymorphism, good reproducibility, extensive genomic coverage, and cost-saving [20]. Moreover, because SSR originates from expressed gene regions and can directly reflect the diversity of related genes, it is widely used in plant genetic breeding, germplasm resource conservation and development, etc. [23,56]. Through SSR analysis of the *R. officinale* transcriptome database, we obtained a large number of unigene sequences containing SSR sites, with an average frequency of 1 SSR/1.1Kb, which is higher than that of species such as coffee (1 SSR/2.16 kb) [57], chickpea (1 SSR/8.66 kb) [58], and *Medicago truncatula* (1 SSR/1.8 kb) [59], but similar to that in *Cassia angustifolia* (1 SSR/1.08 kb) [37]. The SSRs discovered in our work can not only assist the molecular reproduction of *R. officinale*, but also provide more candidate molecular markers to study the genetic variation of *R. officinale* and even *Rheum* species.

## 5. Conclusions

*R. officinale* is a well-known traditional Chinese medicine that is needed for its root and rhizome which have important pharmacological effects. Here, we performed comparative transcriptome analyses on leaves, stems, and roots of *R. officinale*. A total of 236,031 unigenes with N50 of 769 bp was generated, 136,329 (57.76%) of the assembled unigenes were annotated, and 5884 DEGs were identified in the three comparison groups with 175 and 126 key enzyme genes being found in the anthraquinone and catechin/gallic acid biosynthesis pathways, respectively. Interestingly, the phylogenetic analysis of the *PKSIII* superfamily in Polygonaceae indicated only *PL_741 PKSIII1*, *PL_11549 PKSIII5,* and *PL_101745 PKSIII6* of the seven candidate genes probably encoding *PKSIII* in the polyketide pathway, which belonged to the *CHS* group. This valuable genetic information could lay a solid foundation for improving the content of bioactive secondary metabolites in *R. officinale*. Furthermore, the SSRs identified for *R. officinale* would supply substantial genetic molecular markers, together with the transcriptome dataset also providing useful genetic resources for genetic diversity analysis and molecularly assisted breeding at the genomic level.

## Figures and Tables

**Figure 1 genes-13-01592-f001:**
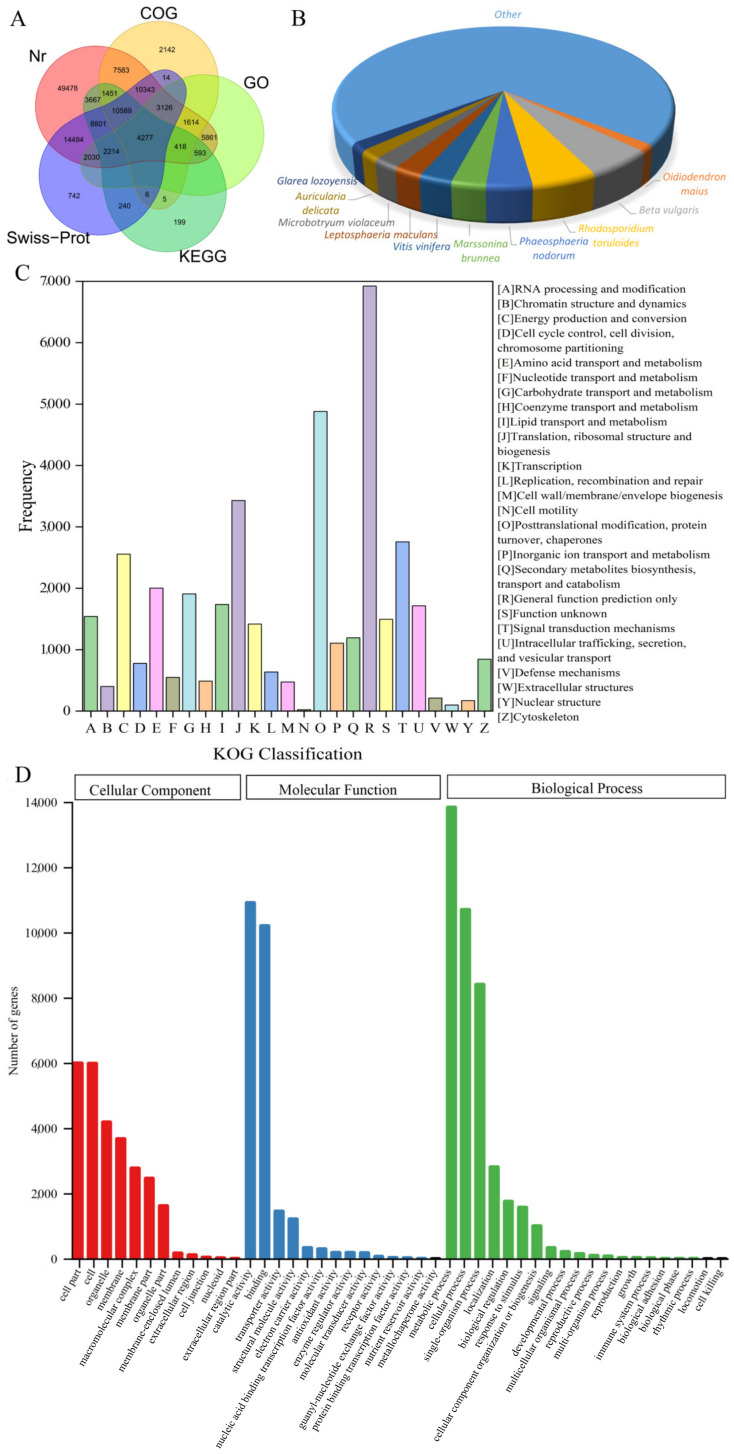
Functional annotation of assembled unigenes. (**A**) Venn diagram for the unigenes annotated by five different databases. The integration of best similarity search results against the NCBI non-redundant protein (Nr), Swiss-Prot, Clusters of Orthologous Groups (COG), Gene Ontology (GO), and Kyoto Encyclopedia of Genes and Genomes (KEGG) databases. (**B**) Species distribution of the annotated unigenes from *R. officinale* in Nr dataset. (**C**) euKaryotic Ortholog Group (KOG) classification. The assembled unigenes were classified into 25 categories in the KOG classification. The *x*-axis indicates the KOG classification, and the *y*-axis indicates the number of unigenes in the category. (**D**) Gene ontology (GO) classification. GO is summarized into three main categories: cellular component, molecular function, and biological process. The *y*-axis indicates the number of unigenes in the category, and the *x*-axis indicates the GO classification.

**Figure 2 genes-13-01592-f002:**
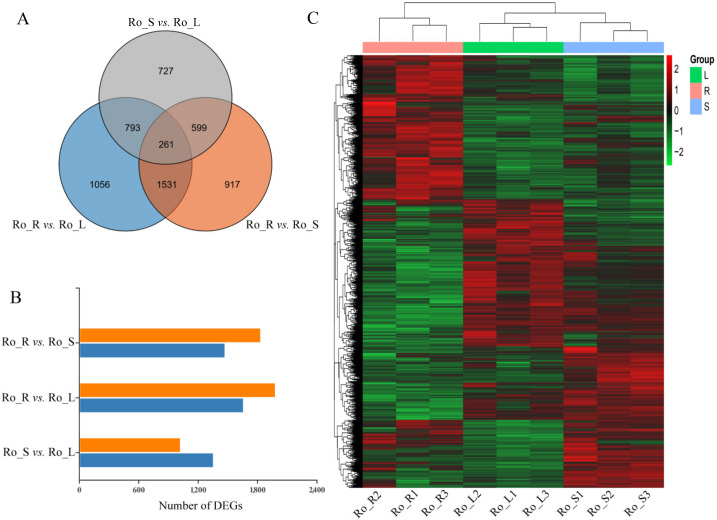
(**A**) Venn diagram of DEGs in the three comparison groups. The three circles represent different comparison groups, and their overlaps indicate the number of common DEGs identified in these comparison groups. (**B**) Number of DEGs in the three comparison groups. The orange rectangles indicate the up-regulated DEGs and the blue rectangles indicate the down-regulated DEGs. (**C**) The hierarchical cluster analysis of DEGs from three comparison groups of different tissues. The rows in the graph represent DEGs, the columns represent samples, and the colors are log-transformed and normalized to the FPKM values. The brighter color represents a higher (red) or lower (green) expression level.

**Figure 3 genes-13-01592-f003:**
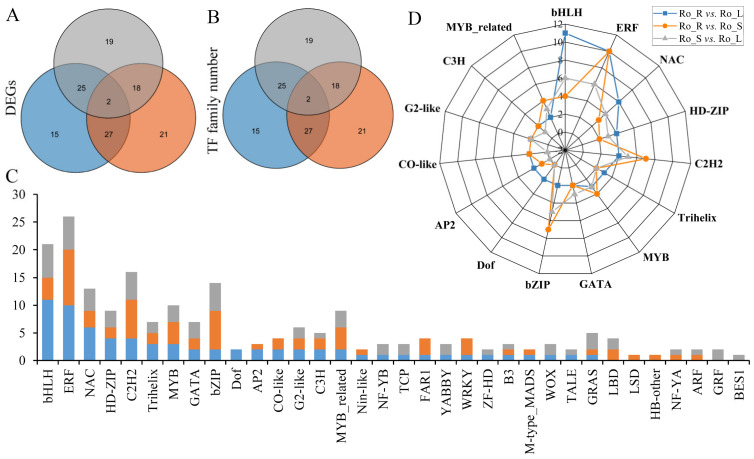
Differentially expressed TFs in the three comparison groups of *R.officinale*. Blue represents Ro_R vs. Ro_L, orange represents Ro_R vs. Ro_S, and gray represents Ro_S vs. Ro_L. (**A**,**B**) represent unigenes and family numbers of differentially expressed TFs in the three comparison groups, respectively. The three circles represent different comparison groups, and their overlaps indicate the number of common DEGs and TF families. (**C**) indicates the number of differentially expressed TFs. (**D**) presents the number of the top 15 differentially expressed TFs in the three comparison groups.

**Figure 4 genes-13-01592-f004:**
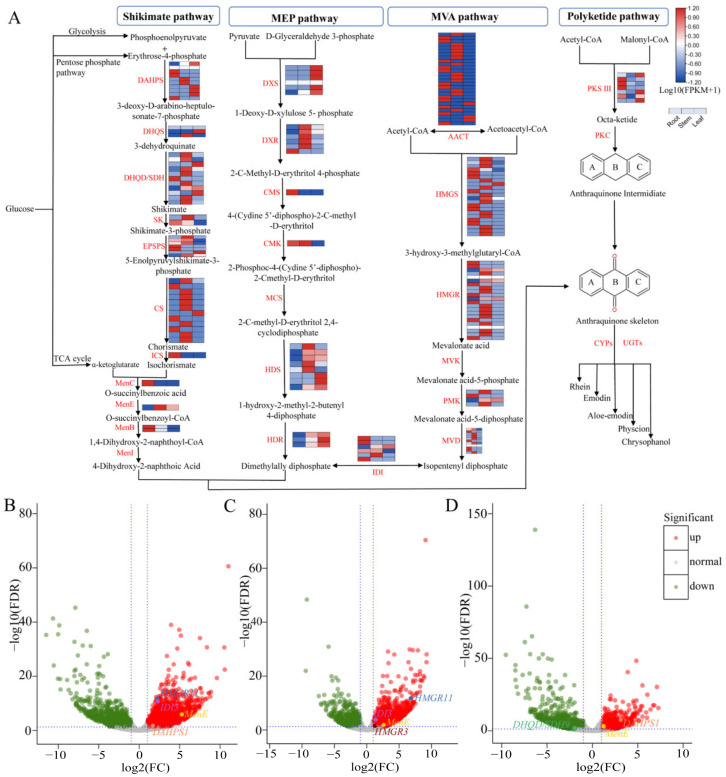
Genes involved in anthraquinone biosynthesis. (**A**) Simplified representation of biosynthetic pathway and expression pattern of unigenes involved in the biosynthesis of anthraquinones. The expression values for each enzyme gene are indicated in color on the log_10_(FPKM + 1) scale for 3 tissues: roots, stems, and leaves. DAHPS: 3-Deoxy-7-phosphoheptulonate synthase; EPSPs: 3-Phosphoshikimate 1-carboxyvinyltransferase; DHQS: 3-Dehydroquinate synthase; DHQD/SDH: 3-Dehydroquinate dehydratase/shikimate dehydrogenase; SK: Shikimate kinase; CS: Chorismate synthase; MenE: o-Succinylbenzoate-CoA ligase; ICS: Isochorismate synthase; MenC: o-Succinylbenzoate synthase; MenB: 1,4-Dihydroxy-2-naphthoyl-CoA synthase; DXS: 1-Deoxy-D-xylulose-5-phosphate synthase; HDR: 4-Hydroxy-3-methylbut-2-en-1-yl diphosphate reductase; DXR: 1-Deoxy-D-xylulose-5-phosphate reductoisomerase; CMS: 2-C-Methyl-D-erythritol 4-phosphate cytidylyltransferase; CMK: 4-(cytidine 5′-diphospho)-2-C-methyl-D-erythritol kinase; HDS: (E)-4-Hydroxy-3-methylbut-2-enyl-diphosphate synthase; IDI: Isopentenyl-diphosphate Delta-isomerase; HMGR: Hydroxymethylglutaryl-CoA reductase. (**B**–**D**) The volcano plots of DEGs in each comparison group are marked with enzyme genes related to the anthraquinone biosynthesis pathway. (**B**) Ro_R vs. Ro_L; (**C**) Ro_R vs. Ro_S; (**D**) Ro_S vs. Ro_L.

**Figure 5 genes-13-01592-f005:**
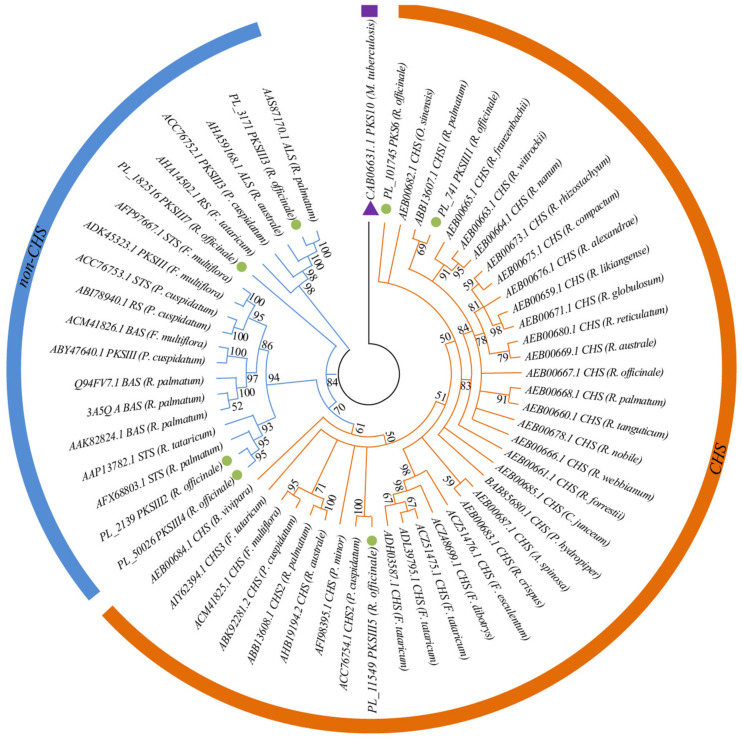
A maximum likelihood phylogenetic tree of the candidate genes in Polygonaceae belonging to *PKS III* superfamily with the bacterial *PKS III* Mycobacterium tuberculosis *PKS10* (purple triangle) set as the outgroup. The sequences shown in the green dots represent the *PKSIII* gene identified in *R. officinale*. The numbers above the branches indicate the bootstrap values for each evolved branch in the tree.

**Figure 6 genes-13-01592-f006:**
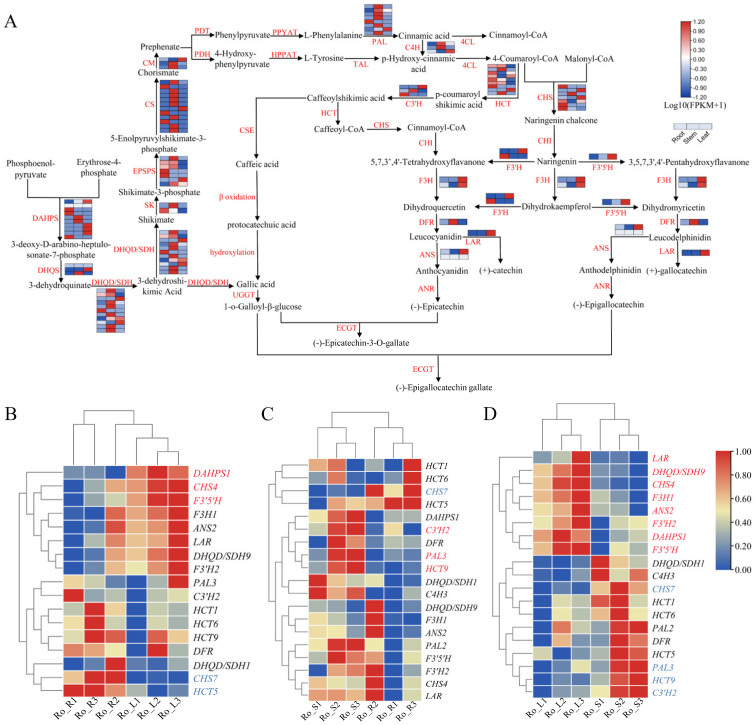
Expression analysis of genes involved in catechin and gallic acid biosynthesis. (**A**) A simplified representation of the catechin and gallic acid biosynthetic pathway. The expression values for each enzyme gene are indicated in color on the log_10_(FPKM + 1) scale for three tissues: roots, stems, and leaves. The deeper color means a higher (red) or lower (blue) expression level for certain genes. (**B**–**D**) Expression of DEGs in catechin and gallic acid biosynthetic pathways. The FPKM value of each DEG was log-transformed and normalized. Genes marked in red represent up-regulated DEGs and genes marked in blue indicate the down-regulated DEGs. LAR: leucoanthocyanidin reductase; UGGT: 1-O-galloyl-β-D- glucosyltransferase; ECGT: O-gallate acyltransferase; CHS: chalcone synthase; CHI: chalcone isomerase; HCT: hydroxycinnamoyl transferase; C3′H: p-coumarate 3-hydroxylase; F3H: flavanone 3-hydroxylase; F3′H: flavonoid 3′-hydroxylase; F3′5′H: flavanoid 3′,5′-hydroxylase; DFR: dihydroflavonol 4-reductase; PAL: phenylalanine ammonia lyase. (**B**–**D**) represent the Ro_R vs. Ro_L, Ro_R vs. Ro_S, and Ro_S vs. Ro_L, respectively.

**Table 1 genes-13-01592-t001:** Summary of the transcriptome assembly results for three tissues of *R. officinale*.

Type	Transcripts	Unigenes
Total reads	1,296,299	443,725
Total number	463,056	236,031
GC content	44.30	45.83
N50 value	1675	769
Min length	179	201
Mean length	839.83	563.98
Max length	24,213	24.213
Sum of lengths	388,889,822	133,117,539

**Table 2 genes-13-01592-t002:** Functional annotation statistics of *R.officinale* unigenes against eight publicly available databases.

Database	Annotated Unigenes	Percentage (%)	300 ≤ Length < 1000	Length ≥ 1000
Nr annotation	126,539	53.61	48,920	20,538
COG annotation	41,568	17.61	15,480	8508
KOG annotation	74,696	31.65	30,882	14,500
GO annotation	20,133	8.53	7357	1952
KEGG annotation	36,789	15.59	14,891	7889
Swiss-Prot annotation	56,876	24.10	23,563	13,690
Pfam annotation	75,256	31.88	28,886	17,752
TrEMBL annotation	102,507	43.43	40,253	28,366
All annotations	136,329	57.76	52,961	21,734

## Data Availability

The data and materials supporting the conclusions of this study are included within the article and its additional files. All the raw reads generated in this study have been deposited in the NCBI with the BioProject accession number of PRJNA827652.

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
