# Peer review of "Comparative Transcriptome Analyses of Different Rheum officinale Tissues Reveal Differentially Expressed Genes Associated with Anthraquinone, Catechin, and Gallic Acid Biosynthesis"

_genes, 2022, doi:10.3390/genes13091592_

Round 1
Reviewer 1 Report
1. In abstract, how can a N50 value less than 1000 i.e., 769 be valid for a comparative transcriptome data?
2. In Line 70-72, you have mentioned that anthraquinone biosynthesis genes have already been reported in the plant, yet why do you want to perform the analysis again?
3. In line 429-430, why have you run the RNA samples on a 2% gel instead of 1% gel?
4. In line 431-432, describe briefly about cDNA library preparation.
5. In line 492, mention the RNA preparation in brief.
6. How did you validate the primers? Mention a paragraph about primer validation.
7. How much time qPCR analysis was repeated for the validation? If it was done only once how can we rely on the given data?
8. You have mentioned about SSR identification, but its validation has been mentioned. Why?
9. In table 1, include the number of reads before and after processing.
10. In all supplementary figures there is no description.
11. In figure S7, why the bar graphs for hexa-nucleotide has been split? The y-axis legends are also unclear. Re-draw the graph properly.
12. In line 137-141, mention briefly about secondary metabolite biosynthesis pathway and their key genes.
13. In line 181, what are eight aforementioned?
14. In line 322, how is it possible to report only 252 Mb of clean data (or) what do you mean by clean data? Why is the N50 value very low?
15. Mention how the statistical analysis was performed for qPCR analysis to determine the error bars.
Reviewer 2 Report
This is an interesting study about the R. officinale transcriptome. Identification the candidate genes associated with anthraquinone, catechin and gallic acid biosynthesis is more useful for the researchers. In addition, its more interesting to see the comparative analysis of various tissues such as root, stem, and leaf in the transcriptome of R. officinale.
Although the research topic and content is more interesting and clear, there are few minor suggestions to improve the content of the manuscript as follows:
1. Abbreviation for KOG is missing. Kindly, incorporate. Also abbreviations of GO, KEGG, COG are provided in Figure 1 heading only. Could you bring it into the manuscript content.
2. In line 78, repeated use of “of” “and” can be modified to improve the quality of the manuscript. Similar changes can be made throughout the manuscript.
3. In line 216, 217 kindly modify the sentence “above-mentioned involved pathways” and improve the clarity for the readers.
4. In line 239, the sentence needs to revised to “Finally, we also found that 112 unigenes were predicted to be …”
5. Please cite tools and software used in this study such as KEGG, Pfam database, bowtie, RSEM, etc.
